# Sulfuration Temperature-Dependent Hydrogen Evolution Performance of CoS$_2$ Nanowires

**Hong-Bo Wang [1,2,*], Zhuo-Jun Qing [1], Hao Zhu [1], Liang Zhou [1,2,*] and Da-Yan Ma [3,*]**

[1] School of Materials Science and Engineering, Chang'an University, Xi'an 710049, China; 2020131016@chd.edu.cn (Z.-J.Q.); 2018231013@chd.edu.cn (H.Z.)

[2] Engineering Research Center of Pavement Materials, Ministry of Education of P.R. China, Chang'an University, Xi'an 710061, China

[3] State Key Laboratory for Mechanical Behavior of Materials, Xi'an Jiaotong University, Xi'an 710049, China

[*] Correspondence: wanghb@chd.edu.cn (H.-B.W.); zhouliang@chd.edu.cn (L.Z.); madayan@mail.xjtu.edu.cn (D.-Y.M.)

**Abstract:** Densely aligned CoS$_2$ nanowires (NWs) on chemically durable stainless steel fibers felt (SSF) substates were synthesized by thermal sulfuring Co$_3$O$_4$ NWs, which were oxidized from hydrothermal synthesized Co(OH)$_y$(CO$_3$)$_{(1−0.5y)}$·nH$_2$O NWs precursors. The effect of sulfuration temperature on the composition, morphology, and HER performance of the products was studied in detail. The results show that the high purity together with the enlarged density of active sites given by the twisted morphology of the CoS$_2$ NWs sulfured at 500 °C guarantee its superior hydrogen evolution reaction (HER) performance compared with other samples sulfured at lower temperatures.

**Keywords:** cobalt disulfide; nanowires; hydrothermal method; hydrogen evolution reaction





## 1. Introduction

Hydrogen (H$_2$), an environmentally friendly energy carrier with high energy density, has been recognized as a promising alternative to traditional fossil fuels to resolve the global warming and energy crisis issues [1–3]. Electrocatalytic water splitting is considered to be a sustainable method for the large-scale production of H$_2$ [4]. To date, the most effective HER catalyst is based on precious metals, especially platinum, which has a very small Tafel slope and high exchange current density [5–7]. However, its high price and low nature storage limit its large-scale application, prompting people to develop other effective, abundant, and excellent stability alternative materials [8–10].

In recent years, pyrite-phase transition metal dichalcogenides (MX$_2$, where typically M = Fe, Co, Mo or Ni, and X = S or Se) have been extensively investigated for HER applications because of their low cost, simple production, and good electrochemical stability [11–14]. Among these MX$_2$ materials, cobalt disulfide (CoS$_2$), a typical Co-based material, has attracted great attention due to its metallic conductivity (6.7 × 10$^3$ S cm$^{−1}$ at 300 K) [15], abundant d orbital electrons, and low H adsorption energy, which affirm it as a promising alternative candidate to replace the costly Ptbased electrocatalysts [16]. In the past few years, CoS$_2$ nanostructures with diverse morphology, such as nanosheets [17], nanowires [18]' nanoneedles [19], nano polyhedrons [20], and nano flowers [21], have been synthesized via hydrothermal process or the thermal sulfuration of Co-based compounds for HER applications. Those works synthetically demonstrated the promising perspective of CoS$_2$ for future HER application. For the synthesis of CoS$_2$ nanostructures, the direct hydrothermal process usually involves problems of impurity incorporation and hybrid products, which hinder the deployment of the inherent superior physical properties of CoS$_2$ [22]. Thermal sulfuration is a versatility approach to synthesize metal sulfides or sulfur-doped compounds with tunable doping concentration by adjusting reaction temperatures or reaction duration [23,24]. The resulting metal sulfides synthesized by

thermal sulfuration usually have relatively high purity because of the controlled vapor environment, and the purity can also be further optimized by adjusting the reaction process conditions. Several studies have shown the advantage of thermal sulfuration to synthesize $CoS_2$ nanostructures with enhanced HER properties by sulfuring Co-based hydroxides [25], cobaltates [26], or polyoxometalates [27].

In this work, densely aligned $CoS_2$ NWs on chemically durable SSF substates were synthesized by thermal sulfuring $Co_3O_4$ NWs. The detailed results show that the composition and morphology of the products strongly depend on the sulfuration temperatures. $CoS_2$ NWs with high purity can be obtained at a sulfuration temperature of 500 °C. The high purity, together with the large density of active sites given by its twisted morphology, guarantee superior HER performance compared with other samples ssulfured at lower temperatures.

## 2. Results and Discussion

As shown in Figure 1a, the SSF substrate used in this work has a large porosity with an average pore size in micron order. The bare SSF fibers have a smooth surface with a uniform diameter of about 10 μm. After the hydrothermal reaction, high-density well aligned $Co(OH)_y(CO_3)_{(1-0.5y)} \cdot nH_2O$ NWs are successfully synthesized on the surface of the SSF substrate with lengths in the range of about 10–15 μm and diameters in the range of 150–200 nm (Figure 1b). After thermal treatment in air, small morphology changes can be found (Figure 1c). Figure 1d–l shows the SEM images of the samples after sulfuration at different temperatures. As is indicated, samples that sulfureted at 300 °C and 400 °C retain the well aligned NW-like morphology (Figure 1d–i), while the sample sulfureted at 500 °C (Figure 1j–l) shows significant morphology changes. The NWs are twisted and intertwined together, forming a reticular structure, leading to an enlarged specific surface area providing more active sites for HER reactions. This obvious morphology change should be related to the accelerated reaction kinetics at high sulfuration temperatures, which results in an obvious anisotropic crystal growth.

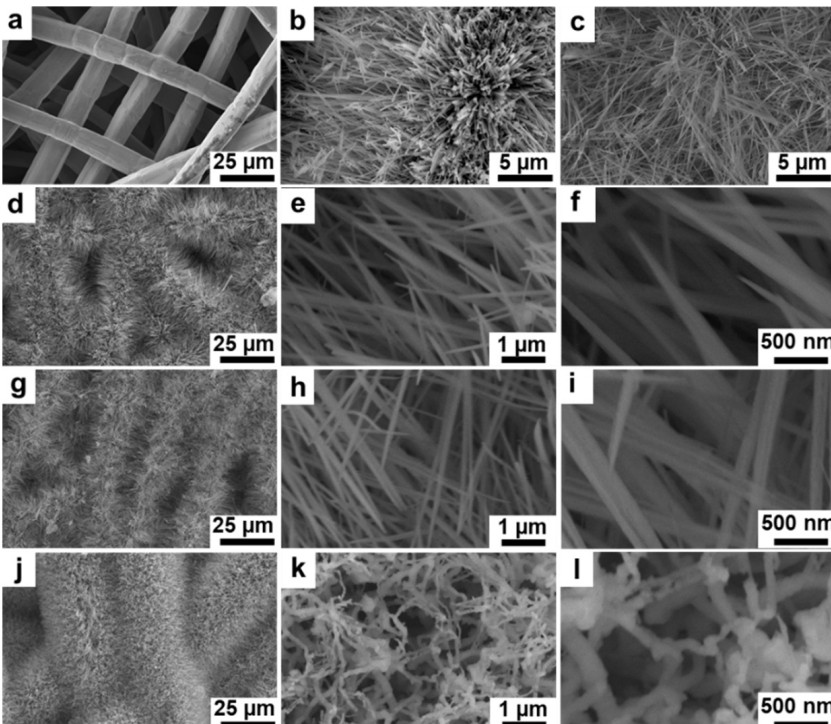

**Figure 1.** The SEM images of (**a**) the SSF substate; (**b**) the $Co(OH)_y(CO_3)_{(1-0.5y)} \cdot nH_2O$ NWs precursors; (**c**) the $CoO_4$ NWs; and the $CoS_2$/SSF samples sulfureted at (**d–f**) 300 °C, (**g–i**) 400 °C, and (**j–l**) 500 °C.

Figure 2a shows the XRD patterns of the samples. Before thermal treatment, the sample only show peaks at 43.5°, 50.7°, and 74.7° attributed to SSF substrate (JCPDS:31-0619). No diffraction peaks from $Co(OH)_y(CO_3)_{(1-0.5y)}.nH_2O$ precursors can be found owing to its low crystallinity degree. The thermal oxidized sample shows diffraction peaks at 31.3°, 36.9°, 38.5°, 44.8°, 55.7°, 59.4°, and 65.2° corresponding to the (220), (311), (222), (400), (422), (511), and (440) crystallographic planes of $Co_3O_4$ (JCPDS:42-1467) [28]. Apart from those peaks, no other impurity peaks can be indicated, confirming the complete oxidation of $Co(OH)_y(CO_3)_{(1-0.5y)}·nH_2O$ precursors after the thermal oxidation process. The sulfured samples show obvious temperature-dependent phase compositions. As can be identified, the diffraction intensity from $CoS_2$ increases with the increase in sulfuration temperature accompanied by the decreased diffraction intensity from $Co_3O_4$. After sulfuration at 500 °C, peaks at 27.9°, 32.3°, 36.2°, 39.8°, 46.3°, 54.9°, and 62.7° are found, attributed to the (111), (200), (210), (211), (220), (311), and (321) crystallographic planes of cubic phase $CoS_2$ (JCPDS 41-1471), respectively [29,30]. The absence of peaks from $Co_3O_4$ indicates the full conversion of $Co_3O_4$ to $CoS_2$. XPS analysis was used to further investigate the chemical bonding states of the sample sulfureted at 500 °C. As shown in Figure 2b, the survey spectrum shows peaks from S 2p and Co 2p, indicating the formation of $CoS_2$. The C 1s and O 1s peaks are mainly attributed to the surface-adsorbed $CO_2$ and surface oxidations. The Fe 2p peak originated from the Fe elements in the substrate. The high-resolution Co 2p XPS spectrum shows two peaks at binding energies of 778.3 and 793.3 eV corresponding to the spin orbitals of Co $2p_{3/2}$ and Co $2p_{3/2}$ from Co-S bond, respectively (Figure 2c) [31]. The satellite peaks at 781.0 and 793.4 eV correspond to Co-O bonds owing to surface oxidations. The peaks at 784.9 and 797.3 eV correspond to the vibronic excitation satellite peaks [32]. The high-resolution S 2p spectrum show peaks at 163.5 eV, 162.7 eV, and 168.9 eV, corresponding to $S_n^{2-}$, $S_2^{2-}$, and surface adsorbed sulfates, respectively [18,33].

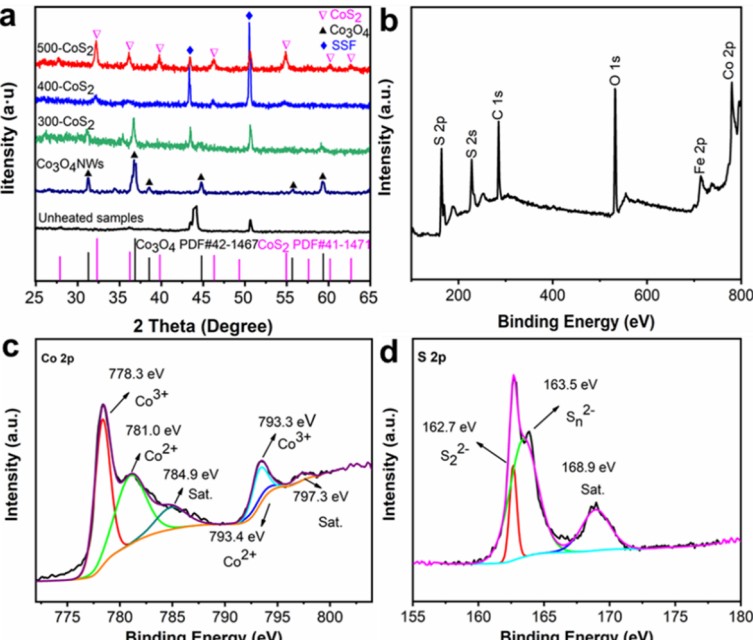

**Figure 2.** (**a**) The XRD diffraction pattern of the untempered by heat treatment, the $Co_3O_4$ NWs samples, and $CoS_2$/SSF; the XPS patterns of the $CoS_2$ NWs: (**b**) the total profile; (**c**) Co 2p; and (**d**) the S 2p profile.

Figure 3 shows the TEM images of the synthesized $CoS_2$ NWs sulfureted at 500 °C. Consistent with the SEM observations, the NWs show a polycrystalline structure with rough surfaces (Figure 3a). The HRTEM image confirms the well-crystallized $CoS_2$ crystallographic feature, where the lattice spacing is measured to about 0.25 nm, well matched with the (210) crystal planes of $CoS_2$ (Figure 3b). The selected area diffraction pattern

confirms the polycrystalline structural feature (Figure 3c). The EDX elemental mapping of $CoS_2$ NWs clearly demonstrates the uniform distribution of Co and S, corroborating the formation of $CoS_2$. A small amount of O element is found to distribute on the surface of $CoS_2$ NWs because of the surface oxidation in the air.

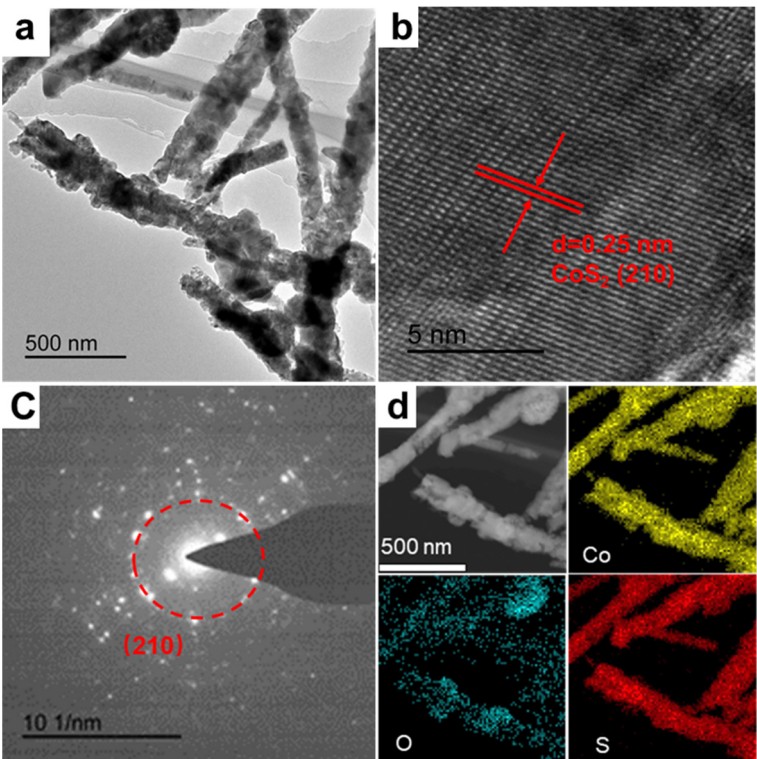

**Figure 3.** (**a**) A TEM image of the $CoS_2$ NWs; (**b**) a high-resolution TEM image; (**c**) the diffraction rings; (**d**) and an elemental distribution map (EDX).

The formation mechanism of the $CoS_2$ NWs can be understood by the chemical reaction shown below that takes place during the synthesis process. In the first step (Equations (1)–(5)), i.e., the hydrothermal synthesis of $Co(OH)_y(CO_3)_{(1-0.5y)}\cdot nH_2O$ precursors, $F^-$ from $NH_4F$ reacts with $Co^{2+}$ to form $[CoF](x-2)-$ ions, and subsequently reacts with $CO_3^{2-}$, $OH^-$, and $H_2O$, resulting in the $Co(OH)_y(CO_3)_{(1-0.5y)}\cdot nH_2O$ precursors. According to previous reports, F- is believed to play a key role that determines the NWs morphology. Firstly, the surface-adsorbed F- can reduce the surface activity of the attached crystal grains, facilitating the growth in the direction perpendicular to the substrate [34]. Secondly, F- can also act as a surface reaction catalyst for the growth of $Co(OH)_y(CO_3)_{(1-0.5y)}\cdot nH_2O$ NWs and thereby maintain its stable growth (Equation (5)) [35]. In the second step, the $Co(OH)_y(CO_3)_{(1-0.5y)}\cdot nH_2O$ NW precursors react with $O_2$ in air at a high temperature to form $Co_3O_4$ NWs (Equation (6)) [36]. Finally, the $Co_3O_4$ NWs can be sulfured to $CoS_2$ in the tube furnace (Equation (7)). At higher sulfuration temperatures, more S gas is generated, and thereby the sulfuration reaction will be accelerated. The grain size of the resulting $CoS_2$ will be enlarged, driven by the accelerated reaction speed and diffusion rate at higher temperatures. The fast anisotropic growth of the $CoS_2$ grains would therefore be the reason for the bent NW morphology.

$$Co^{2+} + xF^- \rightarrow [CoF]^{(x-2)-} \tag{1}$$

$$H_2NCONH_2S + H_2O \rightarrow 2NH_3 + CO_2 \tag{2}$$

$$CO_2 + H_2O \rightarrow CO_3^{2-} + 2H^+ \tag{3}$$

$$NH_3 \cdot H_2O \rightarrow NH^{4+} + OH^- \tag{4}$$

$$[CoF]^{(x-2)-} + (1 - 0.5y)CO_3^{2-} + yOH^- + nH_2O \rightarrow Co(OH)_y(CO_3)_{(1-0.5y)} \cdot nH_2O + xF^- \tag{5}$$

$$Co(OH)_y(CO_3)_{(1-0.5y)} \cdot nH_2O + O_2 \rightarrow 2Co_3O_4 + 3(2n+y)H_2O + (6-3y)CO_2 \tag{6}$$

$$Co_3O_4 + 6S \rightarrow 3CoS_2 + 2O_2 \tag{7}$$

Figure 4 shows the detailed electrochemical HER performances, which are carried out in 1.0 M KOH solution with a three-electrode system. The IR-corrected LSV curves and the corresponding Tafel slopes are shown in Figure 4a,b, respectively. The commercial Pt/C sample shows an overpotential of 0.106 V to drive an 100 mA·cm$^{-2}$ ($\eta$100) and a Tafel slope of 30.28 mV·dec-1, which are consistent with previous reports [37,38]. The $\eta_{100}$ for CoS$_2$ NWs sulfured at 300, 400, and 500 °C are 0.294, 0.279, and 0.224 V, respectively. The $\eta_{10}$ for CoS$_2$ NWs sulfured at 300, 400, 500 °C, and Pt/C are 0.167, 0.152, 0.121, and 0.042 V, respectively. The sample sulfured at 500 °C exhibits the lowest overall potential, indicating its superior HER activity. The sample also shows the lowest Tafel slope (58.15 mV·dec$^{-1}$) among all the sulfured samples, indicating the fastest HER kinetics. Electrochemical impedance spectroscopy (EIS) measurements were conducted to evaluate the charge transfer kinetics at the interface of the cathode surface and the electrolyte. As shown in Figure 4c, the Nyquist impedance spectra of the samples show approximately equal values to R$_s$ and R$_{ct}$, indicating the minimal effect of sulfuration temperature on the charge transfer properties of the samples. The electrochemically active surface areas (ECSA) of the sulfured samples were estimated by electrochemical double-layer capacity (C$_{dl}$). As shown in Figure 4d, the fitted C$_{dl}$ values for samples sulfured at 300, 400, and 500 °C are 9.12, 14.34, and 14.90 mF·cm$^{-2}$, respectively, indicating the largest density of the active sites of the 500 °C sulfured sample.

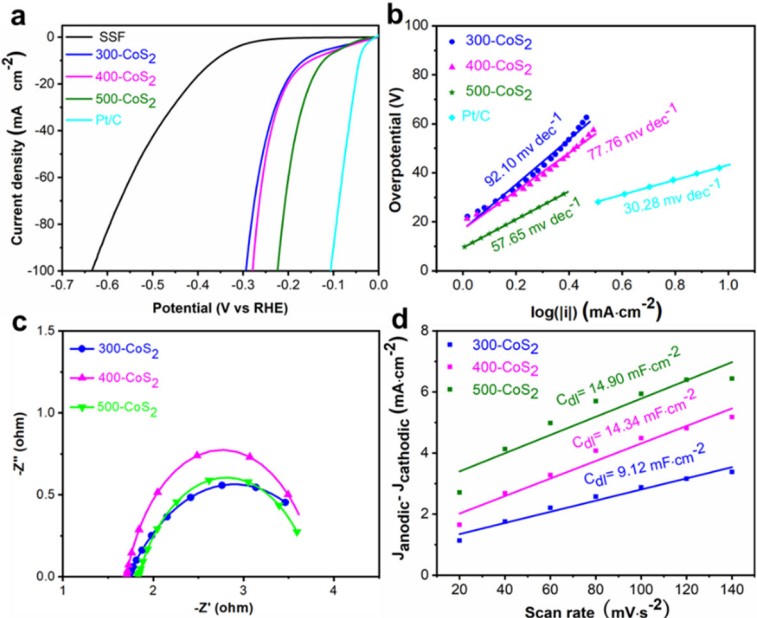

**Figure 4.** CoS$_2$/SSF at different vulcanization temperatures. (**a**) The LSV diagram; (**b**) the Tafel slope diagram; (**c**) the Nyquist diagram; and (**d**) the C$_{dl}$ plots.

The electrocatalytic stability of the sample sulfured at 500 °C was studied by the chronoamperometry and continuous cyclic voltammograms methods. Figure 5a shows a plot of the constant voltage and current density versus time curve. A stable current density is maintained during the potentiostatic measurement for more than 10 h. The polarization curves also show little change after 1000 CV cycles (Figure 5b). Moreover, the sample demonstrates minimal morphology change after the long-term chronoamper-

ometry stability tests. Those results solidify the excellent stability of the 500 °C sulfured $CoS_2$/SSF sample.

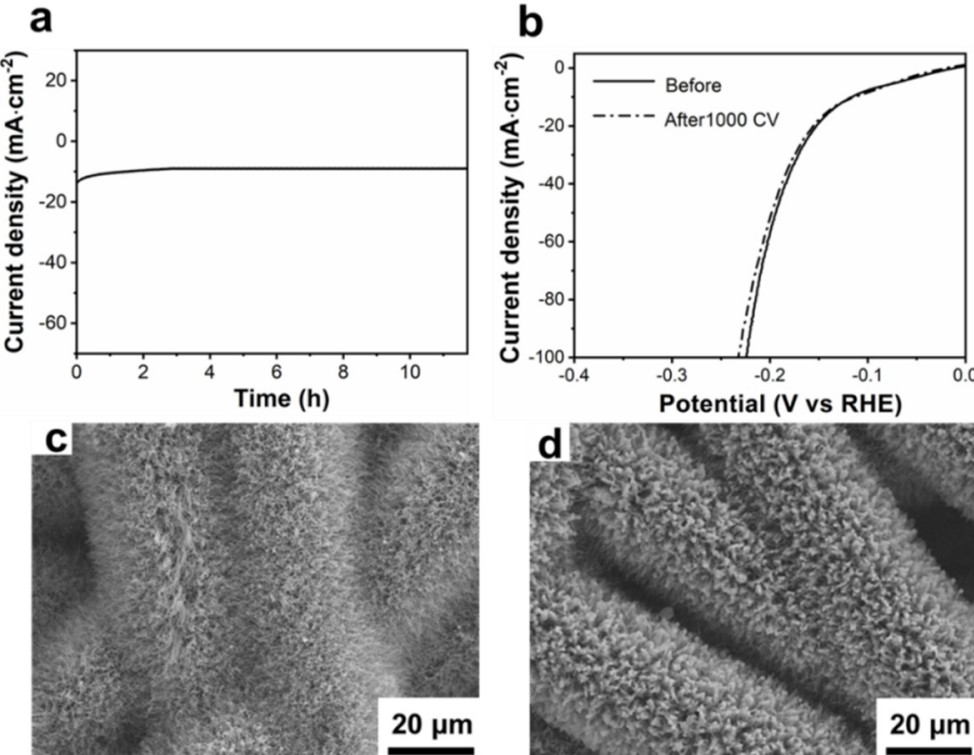

**Figure 5.** The stability tests of the $CoS_2$/SSF sample at sulfured 500 °C. (**a**) The chronoamperometric test result; (**b**) the polarization curves before and after 1000 cycles; (**c,d**) are the SEM images before and after the chronoamperometry testing.

### 3. Experimental

#### 3.1. Materials

Cobalt chloride ($CoCl_2 \cdot 6H_2O$) was purchased from Tianjin Damao Chemical Reagent Factory (Tianjin, China), urea ($CH_4N_2O$) was purchased from Tianjin Hongyan Chemical Reagent Factory (Tianjin, China), and ammonium fluoride ($NH_4F$) was purchased from Guangdong Guanghua Technology Co., Ltd. (Guangdong, China). Deionized water was used as the solvent. All used reagents were of analytical grade and without further purification. 316L SSF felt was purchased from Dingrun Siwang Manufacturing Co., Ltd. (Hengshui, China).

#### 3.2. Synthesis of the Catalysts

The synthesis procedure for the $CoS_2$ NWs is shown in Figure 6. Firstly, SSF felts with sizes of 1 × 3 cm was ultrasonically cleaned by acetone, ethanol, and water for 15 min each, respectively, and then the washed SSF felts were dried in air at 60 °C for 2 h. Secondly, 1.785 g cobalt chloride, 0.601 g urea, and 0.148 g ammonium fluoride were mixed with 40 mL of deionized water and stirred for 10 min to obtain a uniform solution. Then, the solution with the cleaned SSF substrates was transferred into a Teflon-lined autoclave and heated in an oven at 180 °C for 3 h. After the reaction, the autoclave was cooled down to room temperature naturally. The obtained products were washed with acetone, ethanol, and deionized water, respectively, and dried at 60 °C for 30 min, and $Co(OH)_y(CO_3)_{(1-0.5y)} \cdot nH_2O$ NWs on SSF substrate precursors were obtained.

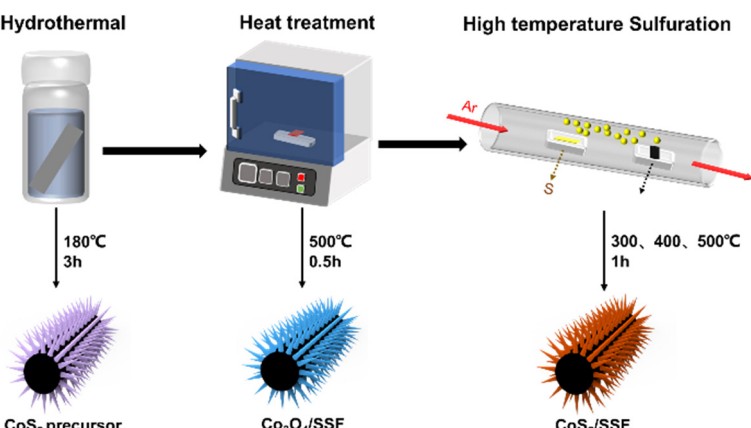

**Figure 6.** A schematic diagram of the preparation process of the $CoS_2$ NWs.

The obtained $CoS_2$ NWs precursor samples were thermal treated in a furnace at 500 °C for 30 min to obtain the $Co_3O_4$/SSF samples. Then, the $Co_3O_4$/SSF samples were sulfured in a tube furnace. Sulfur powder (1 g) and the $Co_3O_4$/SSF samples were put at the upper and lower vent of the tube furnace, respectively. Argon gas was injected into the tube furnace for protection with a gas flow of 10 sccm. The tube furnace was heated to 300 °C, 400 °C, and 500 °C, with a heating rate of 10 °C/min, respectively. After being sulfured for 1 h, the resulting products were obtained.

### 3.3. Characterizations

A Bruker $D_8$ advanced powder X-ray diffractometer (XRD) with Cu Kα radiation was used to identify the crystal structure of the prepared samples (Bruker Group, Karlsruhe, Germany). The morphology of the samples was characterized by S-4800 Hitachi field emission scanning electron microscopy (FESEM) (Tokyo, Japan). The microstructure of the samples was characterized by a JEOL JEM-2100Plus transmission electron microscope (TEM) operating at 200 kV (Beijing, Japan). XPS spectra were measured on a Thermo Scientific K-alpha XPS spectrometer, and the binding energies were corrected by referencing the peak of C1s at 284.60 eV (Thermo Fisher Scientific, Waltham, MA, USA). The contact angle measurement of samples using contact angle gauges (JC2000D1, TEOL, Zhongchen Digital Technology Equipment Co., Ltd., Shanghai, China).

### 3.4. Electrochemical Measurements

Electrochemical measurements were performed with VersaSTAT 3F electrochemical workstation in 1 M KOH aqueous solution (Shaanxi Foreign Trade Import & Export Co., Ltd., Xi'an, China). All electrochemical properties were tested with a typical three-electrode cell at room temperature. The sample, graphite rod, and saturated calomel electrode were used as working, counter, and reference electrodes, respectively. Before the test, the prepared samples were sealed with 704 silica gel to expose a 1 cm$^2$ surface and dried for 12 h. The Pt/C (20%) loaded SSF was also tested for comparison. All the potentials (vs. Hg/HgO) were referenced to a reversible hydrogen electrode (RHE). Linear sweep voltammetry (LSV) was tested at a scan rate of 5 mV/s. The Nyquist plots were measured by electrochemical impedance spectroscopy (EIS) in the frequency range from 100 kHz to 0.01 Hz with an amplitude of 10 mV. Cyclic voltammetry (CV) was used to measure the electric double-layer capacitance at non-faradaic potential and to estimate the effective electrode surface area with scan rates of 20, 40, 60, 100, 120, and 140 mV/s, respectively. The chronoamperometric measurements were used to evaluate the stabilities of the samples. All data were presented after IR compensation to reflect the actual catalytic currents.

## 4. Conclusions

Uniformly distributed $CoS_2$ NWs are successfully prepared on SSF substrate by the hydrothermal synthesis of $Co(OH)_y(CO_3)_{(1-0.5y)} \cdot nH_2O$ NW precursors followed by thermal oxidation and a sulfuration process. After the thermal oxidation, the $Co(OH)_y(CO_3)_{(1-0.5y)} \cdot nH_2O$ NW precursors are converted into $Co_3O_4$ NWs with invisible morphology change. Based on experimental results, the catalytic activity of the $CoS_2/SSF$ electrode can be notably optimized by simply adjusting the sulfuration temperatures. The relatively high-purity $CoS_2$ NWs with significantly bended morphology sulfured at 500 °C offer high conductivity and a large surface area for HER reactions. The overpotential to drive a current density of 100 mA·cm$^{-2}$ is only 0.224 V, which is much lower than that of samples sulfured at lower temperatures.

**Author Contributions:** Conceptualization, H.-B.W.; Validation, H.-B.W. and L.Z.; Investigation, Z.-J.Q., H.Z. and H.-B.W.; Resources, H.-B.W.; Data curation, Z.-J.Q., H.-B.W. and H.Z.; Writing—original draft preparation, Z.-J.Q. and H.Z.; Writing—review and editing, H.-B.W.; Visualization, Z.-J.Q., H.Z. and H.-B.W.; Supervision, L.Z. and H.-B.W.; Project administration, D.-Y.M.; Funding acquisition, H.-B.W. and D.-Y.M. All authors have read and agreed to the published version of the manuscript.

**Funding:** This work was jointly supported by the Natural Science Foundation of Shaanxi Province (No. 2020JM-216), the Key Research and Development project of Shandong Province (No. 2019GGX102023), and the foundation of State Key Laboratory for Mechanical Behavior of Mate-rials in XJTU (No. 20202204).

**Data Availability Statement:** All relevant data are contained in the present manuscript.

**Conflicts of Interest:** The authors declare no conflict of interest.

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
