# Peer review of "Sulfuration Temperature-Dependent Hydrogen Evolution Performance of CoS2 Nanowires"

_catalysts, doi:10.3390/catal12060663_

Round 1

Reviewer 1 Report

This manuscript describes CoS2 NWs on SSF (stainless steel felt) for HER. Based on the demonstration in this work, I would be for the publication of this manuscript in Catalysts after addressing the following.

1) XRD profiles for intermediate phases are missing. They need to be included to reflect the identity of each process.

2) Along with this direction (and, if the sample is Co-LDH after the hydrothermal process), "CoS2 precursor" should be changed to Co-LDH (or better abbreviation).

3) SSF can be dissolved out into the solution to be incorporated into Co-LDH. Not sure if the authors have observed such phenomena. The authors need to scrape off the Co-LDH to carry out the XPS.

4) Tafel region (Figure 5) should be modified. In the current set, 500-CoS2 looks better than Pt/C in terms of the onset potential. 

Author Response

1.       Responses to REVIEWER#1

This manuscript describes CoS2 NWs on SSF (stainless steel felt) for HER. Based on the demonstration in this work, I would be for the publication of this manuscript in Catalysts after addressing the following.

  • XRD profiles for intermediate phases are missing. They need to be included to reflect the identity of each process.

Response from Authors: Thanks for this constructive suggestion. XRD patterns of the unheated sample and thermal treated sample are shown in Fig. 3(a).

  • Along with this direction (and, if the sample is Co-LDH after the hydrothermal process), "CoS2 precursor" should be changed to Co-LDH (or better abbreviation).

Response from Authors: Thanks for this suggestion. "CoS2 precursor" is Co(OH)y(CO3)(1-0.5y)·nH2O NW precursors, LDH is M2+1-x M3+x(OH)2 ]x+[An]x/ n•mHO. So it cannot be abbreviated as Co-LDH.

  • SSF can be dissolved out into the solution to be incorporated into Co-LDH. Not sure if the authors have observed such phenomena. The authors need to scrape off the Co-LDH to carry out the XPS.

Response from Authors: Thanks for your kind work for paper review. The substrate is stainless steel fiber, which has good corrosion resistance and will not react with the solution (the superior corrosion resistance of the SSF substate were verified in our previous work. Journal of Alloys and Compounds, 2021, 876: 160163.).

  • Tafel region (Figure 5) should be modified. In the current set, 500-CoS2 looks better than Pt/C in terms of the onset potential.

Response from Authors: Thanks for this suggestion. The η10 for CoS2 NWs sulfured at 300, 400, 500 ℃ and Pt/C are 0.167, 0.152, 0.121, and 0.042 V, respectively. In comparison, the Pt/C presents the lowest onset potential.

Reviewer 2 Report

The title of the paper "Sulfuration Temperature Dependent Hydrogen Evolution Performance of CoS2 Nanowires" is well presented. The Introduction has latest references up to 2021. As with advancement of technology, references can always be added/update, but for this paper, the list is accepted as it is. Your introduction is well-presented, showing clearly what the authors were prepared to do in their experiments. The materials and methods were researched well, presenting the companies where the materials were obtained and methods that can be easily replicated by others are presented in words and figures.

Figure 2 shows SEM images of the nanowires, but in the caption, section Fig 2(jkl) are not labeled like others from Fig 2(a-i). Please correct this omission in the figure. In the results section, you mention Fig 2(d-i) sulfurazation at different temperatures, while in the caption, you showed temperatures from 2(abc) at 300 degrees C and 2(def) at 400 deg C and (ghi) at 500 deg C. While the results metions briefly how these figures differ, I think you should clarify the figure to be consistent with the text. Also in the text, samples 2(j-k) are sulphurized at 500 deg C, while the figure indicates 2(g-i).

Furthermore, the chemical  information should be refined with difference between reaction and equation in sentences clearly shown: "The formation mechanism of the CoS2 NWs can be understood by the chemical reaction equations shown below that take place during the synthesis process"

I would like to see captions of figures with more description that is found in the text. This applies to all figures. 

Pay attention to references like here " According to previous reports, F - is believed to play a key role that determines the NWs morphology." - add reference to this previous report.

I think omewhere down here, the word kinetic should read kinetics: "The sample also shows the lowest Tafel slope (58.15 mV-dec-1 ) among all the sulfured samples indicating the fastest HER kinetic"

I commend the authors on the science presented in aneasy to understand for reviewers and those who will read the paper.

Author Response

1.       Responses to REVIEWER#2

The title of the paper "Sulfuration Temperature Dependent Hydrogen Evolution Performance of CoS2 Nanowires" is well presented. The Introduction has latest references up to 2021. As with advancement of technology, references can always be added/update, but for this paper, the list is accepted as it is. Your introduction is well-presented, showing clearly what the authors were prepared to do in their experiments. The materials and methods were researched well, presenting the companies where the materials were obtained and methods that can be easily replicated by others are presented in words and figures.

  • Figure 2 shows SEM images of the nanowires, but in the caption, section Fig 2(jkl) are not labeled like others from Fig 2(a-i). Please correct this omission in the figure. In the results section, you mention Fig 2(d-i) sulfurazation at different temperatures, while in the caption, you showed temperatures from 2(abc) at 300 degrees C and 2(def) at 400 deg C and (ghi) at 500 deg C. While the results metions briefly how these figures differ, I think you should clarify the figure to be consistent with the text. Also in the text, samples 2(j-k) are sulphurized at 500 deg C, while the figure indicates 2(g-i).

Response from Authors: We are very sorry for our negligence of the figure labelling. The annotation of the picture has been modified in Figure 2 and is consistent with the text.

  • The chemical information should be refined with difference between reaction and equation in sentences clearly shown: "The formation mechanism of the CoS2 NWs can be understood by the chemical reaction equations shown below that take place during the synthesis process".

Response from Authors: Thanks for this suggestion. "The formation mechanism of the CoS2 NWs can be understood by the chemical reaction equations shown below that take place during the synthesis process" has been changed to "The formation mechanism of the CoS2 NWs can be understood by the chemical reaction shown below that take place during the synthesis process".

  • Pay attention to references like here " According to previous reports, F- is believed to play a key role that determines the NWs morphology." - add reference to this previous report.

Response from Authors: Thank you for your significant reminding. References have been added to the text and marked in red.

[34]         Zeng, S.; Tang, R.; Duan, S.; Li, L.; Liu, C.; Gu, X.; Wang, S.; Sun, D. Kinetically controlled synthesis of bismuth tungstate with different structures by a NH4F assisted hydrothermal method and surface-dependent photocatalytic properties. J. Colloid Interface Sci. 2014, 432, 236-245.

  • I think omewhere down here, the word kinetic should read kinetics: "The sample also shows the lowest Tafel slope (58.15 mV-dec-1) among all the sulfured samples indicating the fastest HER kinetic".

Response from Authors: Thanks for this comment. Modified to "The sample also shows the lowest Tafel slope (58.15 mV-dec-1) among all the sulfured samples indicating the fastest HER kinetics".

Thank you again fou your positive and constructive comments and suggestions on our manuscript.